# AI and Digital Public Infrastructure for Northeast India: A Low-Resource Framework for Inclusive Governance, Education, and Climate Resilience

Ananya Mall[1]

[1] MMVAFP, Research Fellow, Department of Planning & Investment- Papum Pare, India
email@ananyajnv1999@gmail.com

## Abstract

*Northeast India presents a unique technological landscape shaped by geographical isolation, linguistic diversity, ecological vulnerability, and uneven digital infrastructure. The region contains more than 220 languages and dialects and forms part of the Indo-Burma biodiversity hotspot. Despite India's rapid digital transformation, several districts across the Northeast continue to face connectivity gaps, low digital literacy, and limited access to multilingual digital services.*

*This paper explores how artificial intelligence (AI) and Digital Public Infrastructure (DPI) can support governance, education, climate resilience, and public services in Northeast India under low-resource conditions. Drawing from governance implementation experiences in Bihar and administrative observations relevant to Arunachal Pradesh, the study proposes a decentralized AI framework emphasizing multilingual systems, mobile-first access, offline functionality, and community participation.*

*The paper synthesizes recent literature on AI-enabled governance, multilingual educational systems, Digital Public Infrastructure, and climate-sensitive AI applications. It argues that Northeast India can emerge as a model for inclusive, ethical, and low-resource AI innovation.*

**Keywords:** *Northeast India; AI; Digital Public Infrastructure; GovTech; multilingual systems; climate resilience*

## 1. Introduction

Artificial intelligence (AI) is increasingly influencing governance, healthcare, education, agriculture, and environmental management across India. However, most AI systems assume stable internet connectivity, dominant-language interfaces, and digitally mature ecosystems. These assumptions do not fully align with the realities of Northeast India.

The Northeast region comprises eight states with a combined population of nearly 45.7 million people and more than 220 languages and dialects. According to the People's Linguistic Survey of India, many regional languages remain digitally undocumented and underrepresented in AI datasets. The region also forms part of the Indo-Burma biodiversity hotspot and contains nearly 30% of India's biodiversity resources.

Despite progress under Digital India and BharatNet initiatives, connectivity gaps remain significant. Government data from 2025 reported that out of 45,934 villages in the Northeast region, only 40,663 villages had 4G connectivity. Arunachal Pradesh alone had only 3,094 villages connected through reliable 4G services out of 5,993 villages.

These conditions create an urgent need for low-resource AI systems capable of functioning in multilingual and low-connectivity environments.

This paper addresses the following research question:

**How can AI and Digital Public Infrastructure support inclusive governance, education, and climate resilience in Northeast India under low-resource conditions?**

The study combines AI-assisted literature synthesis with governance implementation insights from Bihar and administrative observations relevant to Arunachal Pradesh.

## 2. Methodology

This paper adopts an exploratory qualitative research design supported substantially by AI-assisted synthesis.

The study combines:

1. Comparative governance observations from district-level digital public service implementation in Bihar.
2. Administrative and developmental observations relevant to Arunachal Pradesh.
3. Secondary literature on AI for low-resource regions.
4. Government policy documents related to Digital India, BharatNet, AI for All, and Northeast development.
5. AI-assisted thematic analysis using large language models.
6. The paper does not aim for statistical generalization. Instead, it presents a grounded conceptual framework and implementation-oriented recommendations relevant to low-resource regions.

### 2.1 AI-Assisted Research Process

This paper adopts an exploratory qualitative research design supported substantially through AI-assisted synthesis.

The study combines:

- Secondary literature on AI, GovTech, and Digital Public Infrastructure
- Government reports related to Digital India, BharatNet, and AI for All
- Comparative governance observations from Bihar
- Administrative and developmental observations from Arunachal Pradesh
- AI-assisted thematic analysis using GPT-based systems

The research process involved iterative prompting for:

- Literature synthesis
- Research structuring
- Comparative policy analysis
- Academic language refinement
- Identification of implementation patterns

Additional AI systems including Claude AI and Perplexity AI were used for locating journal references and policy materials, while Napkin AI was used for visual framework generation.

The paper is exploratory in nature and does not claim statistical generalization.

## 3. Findings and Discussion

### 3.1 Multilingual AI and Inclusive Public Services
One of the largest barriers in digital governance across Northeast India is language exclusion. Most digital platforms operate primarily in English or Hindi, limiting accessibility for speakers of regional languages such as Khasi, Mizo, Kokborok, Bodo, Garo, and Nyishi.

Low-resource multilingual AI systems can support:

- Voice-based citizen grievance systems
- AI chatbots for welfare schemes
- Local-language telemedicine systems
- Speech-to-text administrative reporting
- Documentation of endangered languages

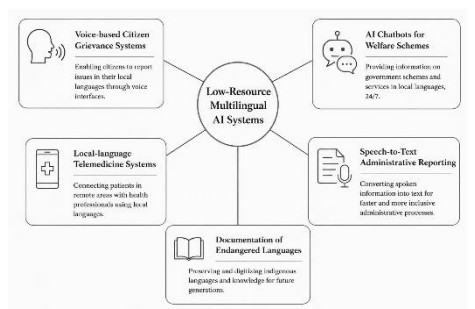

1 Figure:

How Low-resource multilingual AI systems can support Northeast India

Shukla and Shukla (2023) demonstrated how DIKSHA deployed localized multilingual educational content including tribal languages such as Santhali and Ho in Jharkhand. This model provides a replicable framework for Northeast India.

Similarly, the Jugalbandi multilingual chatbot demonstrates how conversational AI can improve access to government schemes in vernacular languages.

These systems should prioritize lightweight AI models capable of functioning under low-bandwidth conditions.

## 3.2 AI-Enabled Education in Remote Areas

Educational inequality remains a major challenge in remote districts across Northeast India because of teacher shortages, transportation barriers, and digital infrastructure limitations.

According to UDISE+ reports, several hilly districts continue to face infrastructural and accessibility challenges affecting educational continuity.

AI-enabled educational systems can improve access through:

- Offline learning applications
- Voice-assisted tutoring systems
- AI-generated multilingual educational content
- Teacher-support platforms

The PM eVIDYA initiative provides an important example of hybrid educational delivery. Television-based educational broadcasting in Bihar helped bridge internet-access gaps during periods of educational disruption.

For Northeast India, similar hybrid systems using television, radio, offline mobile applications, and AI-enabled multilingual learning tools may improve educational accessibility.

## 3.3 GovTech and Digital Public Infrastructure

AI can strengthen governance efficiency in geographically dispersed administrative systems.

Arunachal Pradesh covers approximately 83,743 square kilometers and contains several remote settlements facing transportation and communication barriers during monsoon periods.

Districts such as Upper Subansiri and Kurung Kumey frequently experience landslides and road disruptions affecting healthcare delivery, educational outreach, and emergency coordination.

AI-assisted GovTech systems could support:

- Automated district reporting
- Mobile grievance systems
- Voice-based frontline worker reporting
- Predictive infrastructure-risk identification
- AI-assisted planning dashboards

Comparative implementation experiences from Bihar's digital health systems show that frontline workers often struggle with repetitive data entry, reporting delays, and limited digital training.

Bhagawati (2020), in a study on e-governance systems in Assam and Nagaland, identified weak vertical integration, lack of ICT training, and infrastructural limitations at district and block levels as major barriers to effective governance.

Recent literature on Digital Public Infrastructure (DPI) provides additional conceptual grounding for these systems.

Chawala and Iyer (2025) describe India Stack through three interoperable layers:

| Layer | Component | Purpose |
|---|---|---|
| Identity | Aadhaar | Authentication and identity |
| Payments | UPI | Interoperable digital payments |
| Data | Account Aggregator | Consent-based data sharing |

Such interoperable and low-cost infrastructure models are highly relevant for low-resource AI systems in Northeast India.

## 3.4 Climate Resilience and Ecological Monitoring

Northeast India is highly vulnerable to floods, landslides, soil erosion, and climate variability.

According to the National Disaster Management Authority (NDMA), states such as Assam and Arunachal Pradesh experience recurring flood and landslide events during monsoon periods.

AI-enabled geospatial systems can support:

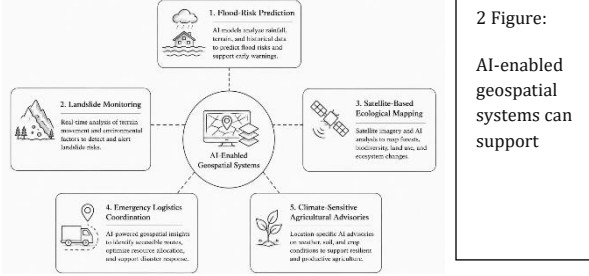

2 Figure:

AI-enabled geospatial systems can support

- Flood-risk prediction
- Landslide monitoring
- Satellite-based ecological mapping
- Emergency logistics coordination
- Climate-sensitive agricultural advisories

For example, AI-supported multilingual mobile alert systems could help communities along the Siang and Subansiri river basins receive localized flood-risk notifications. An important opportunity also lies in integrating traditional ecological knowledge into climate-resilience systems. Indigenous communities across Northeast India possess extensive knowledge related to river behavior, biodiversity cycles, medicinal plants, and environmental change. AI systems designed with community participation and ethical safeguards can help preserve and utilize this knowledge responsibly.

## 3.5 Key Research Gap

Existing literature on AI and Digital Public Infrastructure primarily focuses on urban or nationally scalable systems. Few studies systematically combine:

- Low-resource AI systems
- Digital Public Infrastructure
- Climate resilience applications
- Multilingual governance systems
- Community-centered implementation

specifically for Northeast India.

This paper contributes toward addressing this gap by proposing a decentralized and multilingual AI framework tailored to the ecological, linguistic, and infrastructural realities of the region.

## 4. Conclusion

AI and Digital Public Infrastructure hold significant potential for transforming governance, education, public services, and climate resilience in Northeast India.

However, meaningful implementation requires moving beyond centralized and language-exclusive technology models.

This paper argues that effective AI systems for Northeast India should:

- Function in low-connectivity environments
- Support regional languages
- Operate through mobile-first systems
- Include community participation
- Protect indigenous knowledge systems
- Use lightweight and low-cost infrastructure

Northeast India should not merely be viewed as a deployment region for existing technologies. Instead, it can emerge as a global model for inclusive, ethical, and low-resource AI innovation.

## 5. Figures & Tables

- 1 Figure:  How Low-resource multilingual AI systems can support Northeast India
- 2 Figure: AI-enabled geospatial systems can support
- 1 Table: India Stack through three interoperable layers in Digital Public Infrastructure (DPI)

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

Estimated AI contribution to the overall paper: approximately 80%.

---

**AI Involvement Checklist**

| Parts of the Research | Score (1–4) |
|---|---|
| Idea generation | 3 |
| Literature selection | 3.5 |
| Literature review | 4 |
| Generation of research questions | 3 |
| Research design and thematic structuring | 3.5 |
| Data collection and reference identification | 3 |
| Data analysis and interpretation | 3 |
| Visualization and framework creation | 4 |
| Writing and language refinement | 4 |
| Human contextual validation | 2.5 |
| **verage score** | 3.3 |