# OpenReview forum: "AI and Digital Public Infrastructure for Northeast India: A Low Resource Framework for Inclusive Governance, Education, and Climate Resilience"
_NortheastGenAI/2026/Workshop — NortheastGenAI 2026 Workshop Submission_

### Official Review · ~Badal_Nyalang1 · 2026-05-23
**Honest disclosure saves it — Weak Accept**

**Rating:** 6
**Confidence:** 4

**Review:**

**Relevance: Good**
Fits T3 cleanly. The multilingual governance + climate resilience angle is appropriate, and the NE India focus is consistent throughout.

**Plausibility: Weak**
This is the core problem. The Bihar governance observations are never actually described — they're just asserted as comparative evidence. The Arunachal Pradesh grounding is similarly thin: general facts about landslides and district sizes, nothing original. The "decentralized AI framework" promised in the abstract never actually materialises as a framework — it's a bullet list.

The 80% AI contribution score is honest, but it shows. The paper reads as a synthesis of things any LLM would say about low-resource AI in underserved regions. Nothing is grounded in actual field data, interviews, or even a structured literature review.

**Novelty: Weak**
Everything here is already well-known in the DPI and low-resource AI literature. The research gap claim in §3.5 is overstated.

**Clarity: Moderate**
Readable and well-formatted. The methodology section is duplicated almost verbatim (§2 and §2.1 overlap significantly). The figures are AI-generated and add little analytical value.

**Verdict: Weak Accept**
Honest disclosure saves it. But the paper is largely an AI-generated literature summary dressed as a framework paper. Recommend the author ground it in at least one concrete implementation observation before presentation.

*This review was generated with AI assistance and checked by the workshop chairs.*

---

### Decision · Program_Chairs · 2026-05-23

**Decision:**

Accept (Weak)

**Comment:**

This paper addresses a relevant and underexplored area: AI-ready digital public infrastructure for Northeast India, with a consistent regional focus. The multilingual governance and climate resilience framing fits the workshop well. The core weakness is that the proposed framework remains a high-level synthesis without original field data or implementation evidence. The methodology section also contains a near-verbatim duplication that should be addressed if possible before presentation.

Decision: Accept (Weak)